# Headache in Multiple Sclerosis: A Narrative Review

**DOI:** 10.3390/medicina60040572

**Published:** 2024-03-30

**Authors:** Bożena Adamczyk, Natalia Morawiec, Sylwia Boczek, Karol Dańda, Mikołaj Herba, Aleksandra Spyra, Agata Sowa, Jarosław Szczygieł, Monika Adamczyk-Sowa

**Affiliations:** Department of Neurology, Faculty of Medical Sciences in Zabrze, Medical University of Silesia in Katowice, ul. 3 Maja 13-15, 41-800 Zabrze, Poland; sylwiaboczek10@gmail.com (S.B.); karoldanda@gmail.com (K.D.); s78446@365.sum.edu.pl (M.H.); aleksandra1828@gmail.com (A.S.); s91753@365.eum.edu.pl (A.S.); jwszczygiel@gmail.com (J.S.); m.adamczyk.sowa@gmail.com (M.A.-S.)

**Keywords:** multiple sclerosis, headache, prodromal syndrome, migraine, tension-type headache

## Abstract

*Background*: Multiple sclerosis (MS) is a chronic inflammatory demyelinating disorder of the central nervous system characterized by autoimmune-mediated damage to oligodendrocytes and subsequent myelin destruction. *Clinical implications*: Clinically, the disease presents with many symptoms, often evolving over time. The insidious onset of MS often manifests with non-specific symptoms (prodromal phase), which may precede a clinical diagnosis by several years. Among them, headache is a prominent early indicator, affecting a significant number of MS patients (50–60%). *Results*: Headache manifests as migraine or tension-type headache with a clear female predilection (female-male ratio 2-3:1). Additionally, some disease-modifying therapies in MS can also induce headache. For instance, teriflunomide, interferons, ponesimod, alemtuzumab and cladribine are associated with an increased incidence of headache. *Conclusions*: The present review analyzed the literature data on the relationship between headache and MS to provide clinicians with valuable insights for optimized patient management and the therapeutic decision-making process.

## 1. Introduction

### 1.1. Epidemiology

Multiple sclerosis (MS) is a chronic neuroinflammatory autoimmune disease affecting the central nervous system (CNS). It is estimated that approximately 51,000 individuals in Poland are affected by MS with a prevalence of approximately 134 cases per 100,000 individuals. Globally, the number of MS patients increased from 2.3 million people in 2013 to 2.9 million in 2023. Worldwide, 1 in 3000 people suffer from MS, and in countries with a higher incidence of disease, even 1 in 300 people are diagnosed with MS [1]. The disease progression is difficult to predict. In some individuals, MS is characterized by periods of relapse and remission while for others it has a progressive pattern [2].

### 1.2. Phenotypes and First Symptoms of MS

MS can manifest through different patterns, of which relapsing–remitting MS (RRMS) is the most common phenotype. It is characterized by exacerbations of symptoms (relapses) followed by complete or partial recovery (remission), often resulting in mild to moderate disability. Approximately 10–15% of patients may develop primary-progressive MS (PPMS). It is defined by a progressive decline in neurological function without remission. Clinically isolated syndrome (CIS) is the initial manifestation of MS symptoms prior to a confirmed MS diagnosis. Secondary progressive MS (SPMS) is characterized by the gradual progression of the disease following an initial relapse, with some but not all patients with RRMS progressing to SPMS [3].

### 1.3. First Symptoms of MS

MS may induce various neurological symptoms. Over the last decade, accumulating evidence has suggested the occurrence of non-specific symptoms several years prior to the diagnosis as part of the prodromal syndrome. Fatigue, cognitive impairment, pain, migraine, gastrointestinal complaints, bladder disturbances, sleep disorders, anemia, depression, and anxiety are common symptoms that manifest during the MS prodrome [4,5,6,7,8]. Recent research has identified headache as a crucial symptom of the prodromal phase, potentially indicating disease development many years before the initial demyelinating event. Furthermore, headache may frequently persist even after diagnosis and treatment administration or may arise de novo as a side effect of therapy [9].

This review investigated headache as an indicative factor prompting the initiation of a diagnostic process for MS. It outlined the most frequent headache types in patients with MS. Additionally, the review demonstrated the associations between immunomodulatory therapies and the occurrence of headache. Disease-modifying therapies (DMTs) may cause headaches or worsen existing ones. The use of interferon-beta, laquinimod, fingolimod, or cladribine has been proven to cause or exacerbate headaches [10,11,12]. 

The increased incidence of headaches in people with MS raises questions regarding its significance as an important manifestation of the illness. Hence, it remains uncertain whether young individuals who exhibit typical or atypical headache phenotypes ought to undergo an MRI to obtain an early diagnosis of multiple sclerosis. We have decided to investigate the relationship between MS, its treatment, and headache.

## 2. Materials and Methods

We searched PubMed, PMC, Google Scholar, and SCOPUS databases. To find relevant data concerning our topic, we chose to use the following search terms—“multiple sclerosis prodrome”, “MS prodrome”, “multiple sclerosis headache”, “multiple sclerosis migraine”, “cortical spreading depression”, “migraine demyelination”. Additionally, the references described in the selected studies were screened for potentially useful material. To obtain a wider understanding of the subject and comprise a significant amount of research data, we included a variety of types of studies—meta-analyses, case-control studies, literature reviews, cross-sectional studies, and prospective, longitudinal studies. The collected data are presented in the form of a table (Table 1) as well as discussed in the preceding paragraphs.

## 3. Multiple Sclerosis Prodrome

Multiple sclerosis prodrome is a set of early signs or symptoms that occur before the start of classical disease, prior to a final diagnosis being made [8]. It can manifest several years before the development of the first demyelinating event. Evidence indicates that MS was associated with significantly higher annual healthcare utilization 1–5 years before diagnosis compared to the control group, which suggests the existence of a preclinical phase of MS [5]. Furthermore, prodromal manifestations of RRMS and PPMS may vary. Patients with PPMS had more nervous system-related encounters compared to subjects with RRMS [6].

Based on case-control studies, the most commonly reported issues by patients during the prodromal phase include gastric, intestinal, urinary, anorectal, and dermatological disorders, anemia, anxiety, depression, insomnia, fatigue, cognitive impairment, headache, and various types of pain [6,7,8]. Many of the MS prodromal traits identified thus far are non-specific and prevalent in the general population; no single feature is adequate to distinguish an individual with prodromal MS. While knowledge of a prodromal phase in MS is still in its early stages, it is crucial to determine how to best proceed from quickly expanding data to research-related action [8]. Immediate implications include refining the concept of the MS continuum to include a prodromal phase. This will help to determine the true “at risk” timeframe when considering exposures that may cause MS. Major long-term consequences include earlier MS detection, improved prognosis as a result of early disease management, and the prospect of MS disease prevention [8]. However, no formal criteria to define or diagnose prodromal MS have been defined.

Efforts are currently underway to develop formal criteria for diagnosing the prodromal phase of MS, but this is difficult because many of the symptoms of this phase also occur in people who do not suffer from MS. Since the symptoms of the disease are non-specific, new biomarkers are being sought to enable the earliest possible detection of the disease. One of them is RIS (radiologically isolated syndrome). We describe RIS when the patient’s symptoms are not characteristic of MS, or when the patient has no symptoms, but MRI results suggest the diagnosis of MS [8,13]. Tremlett et al. showed that patients presenting with radiologically isolated syndrome (RIS) could simultaneously report clinical symptoms, which indicates that they were found in the preclinical phase [8]. Additionally, RIS could serve as a marker of the MS prodrome. Another marker tested to detect prodromal MS is advanced neuroimaging—both patients in the prodromal phase and those with developed MS show reduced volume of the cerebellum, thalamus, and the entire brain compared to the control group. CSF can also be examined for the presence of oligoclonal bands or elevated levels of NfL (neurofilament light chain) [8]. 

## 4. Headache as the First Manifestation of MS

In their prospective longitudinal study, Gebhardt et al. found that headache in early MS was a crucial symptom. They noted a significant decrease in the occurrence of headache within six months after the initial clinical manifestation of MS [14]. Another study, including a cohort of 451 patients with RIS, showed that headache prompted brain MRI examination, which resulted in the diagnosis of RIS in 42.1% of patients (*n* = 190). Although that study did not identify headache as a significant risk factor for developing a clinical episode, it remained the most prevalent symptom prompting brain MRI investigation [15]. Similar findings were found in another cohort study involving 44 patients, in which migraine headache was the main reason for brain MRI (38.6%). However, it was not identified as a risk factor for developing MS [13].

Vacca et al. examined 238 patients with MS, of whom 212 presented with a history of headache. Almost 70% of patients (*n* = 85) reported headache preceding the onset of MS [16]. Another study showed the occurrence of headache prior to the initiation of MS therapy in 20% of patients with this symptom. Other subjects presented with headache after therapy and 101 of 515 reported it as an early symptom [10]. A similar frequency was observed in a study by Doi et al. in which 22% of patients experienced headache in the prodromal phase [17]. Rolak et al. demonstrated that 13% of MS patients with headache presented with this symptom at the onset of the disease compared to 7% from the whole cohort of MS patients (*n* = 104). Although it could be a coincidence, each patient met the 2017 McDonald criteria confirming the diagnosis [18]. Mijajlovic et al. described a case report of a 45-year-old male patient with cluster headache as the initial and only clinical manifestation of MS, which was diagnosed after cerebrospinal fluid isoelectric focusing and brain MRI investigation. Multiple demyelinating lesions were found in the brainstem [19], which is a typical location in MS patients with accompanying cluster headache [20].

Rościszewska-Żukowska et al. collected data from 419 consecutive patients with RRMS. All patients were diagnosed according to McDonald’s 2017 criteria and evaluated in an outpatient MS clinic. Each patient was examined by a questionnaire. The first part was used to collect demographic and clinical MS data, while the second part checked for primary headache criteria and included clinical symptoms and headache history. Primary headaches were confirmed by a neurologist and headache specialist using the 2018 ICHD-3 headache classification criteria. According to these criteria, patients were classified with various types of headaches. In this cross-sectional study, the authors found that a remarkably high percentage of patients (78.8%) reported the onset of headache before MS. Furthermore, only one-third of all patients with headache presented with more frequent pain attacks after the diagnosis of MS [21].

Headache as a prodromal symptom of MS is reported more frequently in women (39.7%) than in men (26.5%). Also, individuals with the highest annual EDSS increase (0.35) reported significantly more headache episodes. According to Kania et al., the occurrence of the prodromal phase was independent of age at disease onset [9]. 

Various studies have shown that headache can be a common symptom in the prodromal and early phases of MS. Furthermore, the association between headache and diagnostic procedures, including brain MRI, underscores the potential significance of headache as an indicator in the clinical assessment of MS.

## 5. Headache in the Course of MS

Headache can occur after the diagnosis of MS and may show different trajectories. As a result, increasing or decreasing headache is reported, depending on therapeutic interventions. A meta-analysis based on 16 studies showed the pooled estimated prevalence of primary headache in MS at 56% [22]. Such headache could manifest during the disease course, relapses, and several years preceding the diagnosis.

Polish researchers observed that 50% of MS patients in their study group (*n* = 144) presented with some form of head-related pain (65.3% of patients reported a history of pain). Of note, headache before the onset of MS was described as more intense [23].

Various studies showed that migraine or tension-type headache (TTH) are the predominant headache types in MS patients with an incidence of approximately 2.6 times higher compared to the general population [24,25,26]. Headache significantly influences quality of life, with a higher prevalence in females with MS. Beckmann and Türe found that 68% of MS patients presented with headache and 80% of patients reported headache after the onset of treatment. The study found that migraine was the most common type of headache (17% of patients without aura and 7% of subjects with aura). Approximately one-fifth of patients developed tension headache, while 38% had medication-overuse headache and 2% had unclassified headache [10]. In a meta-analysis by Mohammadi et al., 24% of patients with MS experienced migraine. The research, however, neglected the impact of age, the duration of MS, or migraine diagnosis techniques. Additionally, patients with MS showed a significantly higher risk of migraine [26].

Togha et al. found that headache was significantly more prevalent in the relapse phase in almost 50% of patients (45.6%) compared to the remission phase (38.6%). The connection between gender and the incidence of headache in relapse and remission was not reported in the study. Compared to the control group, pain was reported much more frequently during the relapse phase. The severity of pain did not significantly correlate when controls and patients in remission and relapse were compared [27]. During remission and relapse, migraine was the most frequently reported form of pain, followed by TTH. The fronto-orbital region was the location of headache in most patients, while photophobia and phonophobia were the most prevalent concomitant symptoms. Pain was mostly assessed as moderate during the three months following admission for attack-period treatment. It was most frequently rated as severe (22.8%) and compressing (28.1%) during the remission period. Furthermore, headache occurred much more frequently in patients in the relapse phase having MS < 3 years compared to relapse subjects with more than three years of MS. However, no such association was noted during the remission phase [27].

### 5.1. Quality of Life

Husain et al. stressed a higher prevalence of depression in patients with both migraine and MS, highlighting the importance of screening and the management of the comorbidity [28]. Based on the 12-Item Short Form (SF-12) questionnaire, individuals with headache showed significantly lower psychological quality of life compared to those without headache, and they reported more frequent cognitive fatigue. These findings indicate a decreased quality of life related to headache [29]. The high prevalence of fatigue (89%) in 100 MS patients shows that special attention should be paid to fatigue in the treatment plan for MS patients. As a result, questions regarding fatigue should be included in the medical history of MS patients [30,31]. 

Villani et al. discovered that MS patients with concomitant migraine scored similarly to those without in both physical health composite score (PH-CS) and mental health composite score (ME-CS). In contrast, they found that migrainous MS patients scored lower on the prodromal phase (PP), bodily pain (BP), and health perception (HP) subscores. The remaining Multiple Sclerosis Quality of Life-54 (MSQoL-54) subscales revealed no significant difference between the two groups. Although the multivariate models indicated that migraine did not alter either the PH-CS or the ME-CS, the presence of concomitant migraine was an independent predictor of limitations due to physical problems (RL-P) and HP. Finally, they observed that the Migraine Disability Assessment Test (MIDAS) score was substantially connected to MSQOL-54 subscales affected by migraine: RL-P, BP, and HP. In contrast, migraine frequency was exclusively associated with RL-P and neither BP nor HP [32].

### 5.2. Gender

Beckmann et al. studied 782 individuals diagnosed with RRMS and primary headaches using the IHS classification. At least two neurologists examined each patient and took a complete neurological history, including a history of headaches. During the initial appointment, the patient filled out a headache questionnaire. The research excluded people with sporadic headaches as well as those with other conditions that could potentially induce headaches [10]. The study showed the incidence of headache in MS patients to be two to three times higher in females compared to males [21,28]. Migraine, particularly without aura, was the most frequently reported type of headache in female and male MS patients. Other headache types were reported with similar frequency in both genders [10].

### 5.3. Age

A statistically significant age difference was observed between MS patients with and without headache. The mean age of subjects without headache was 33.7 compared to 37.1 in patients with headache [10].

## 6. Headache Types in MS

### 6.1. Migraine Headache

Migraine headache (MH) is a genetically influenced complex disorder characterized by episodes of moderate-to-severe headache, which are usually unilateral and are accompanied by nausea and increased sensitivity to light and sound. The International Headache Society’s headache classification committee classifies migraines into subtypes: migraine without aura, migraine with aura, and chronic migraine [33]. Migraine is reported to be the most prevalent type of headache in patients with MS in many studies [10,14,16,21,34,35]. It occurs at least twice as often in patients with MS than in healthy individuals [36]. The estimated global prevalence of migraine is 14% [37]. There are many similarities between migraine and MS [36]. Both diseases mainly affect young individuals, with female predominance and a higher incidence in Caucasian populations, which indicates a relationship between these diseases [36].

There are several theories explaining the mechanism of the frequent co-existence of MS and migraine (Figure 1). In animal models, the process of demyelination of the cortical myelin sheath has been identified as a crucial factor in the pathogenesis of migraines. This is also the case for the pathogenesis of MS [38]. Demyelination and the neurodegeneration of the brainstem, periaqueductal gray matter (PAG), trigeminal nucleus, thalamus, somatosensory cortex, and other pain-sensing areas cause migraine attacks by activating the pathophysiological cascade [36]. The midbrain comprises three significant structures that are implicated in the pathogenesis of migraine headaches, namely the primary center for serotoninergic cells, the major noradrenergic center, and the PAG, a complex structure that transmits information to and from higher cortical areas and the spinal cord. The rostral ventromedial medulla and the dorsolateral pontomesencephalic tegmentum are intimately connected to each other, reducing the firing of nociresponsive neurons in the dorsal horn. Chemical stimulation of the PAG induces defense responses and modifications in sensory processing, as well as alterations in cerebral blood flow [39]. During migraine with aura in MS, depression of the cortex may result in increased blood–brain barrier permeability, which may predispose the myelin of neurons to T-cell infiltration. This approach suggests a possible connection between migraines, blood–brain barrier (BBB) integrity, and the infiltration of immune cells into myelin [40]. An imaging study by Gee et al. found a much higher likelihood of migraine-like headache in patients with lesions in the PAG (OR 3.91, 95% CI, *p* < 0.0001) as compared to MS patients without plaques in the PAG [39]. In the case of TTH, there was a 2.5-fold increase (OR 2.58) and a 2.7-fold increase (OR 2.77) in the combination of migraine and TTH [39]. Furthermore, abnormal excitability of the cerebral cortex in patients with cortical MS lesions (found in the earliest stages of MS and positively related to the severity of physical and cognitive impairments) leads to greater susceptibility to cortical spreading depression (CSD), which is considered the leading cause of migraine, mainly with aura [41]. CSD, a gradual depolarization wave that is followed by a suppression of brain activity, is a highly intricate phenomenon that involves significant changes in neural and vascular function [42]. Synaptic activity, extracellular ion concentrations, blood flow, and metabolism are significantly affected [36]. The augmentation of atypical excitability in the cortex resulting from cortical demyelination and impaired remyelination in MS increases the susceptibility to widespread depression, which is likely the underlying cause of migraine with aura [43]. Another theory is based on immunological changes. An intense immune response in MS patients increases the number of activated T-cells and B-cells in the meninges. These cells express higher levels of calcitonin gene-related peptide (CGRP), which is important in the pathogenesis of migraine. In response to the action of CGRP, mast cells in the dura mater release histamine, serotonin, heparin, tryptase, and proinflammatory or anti-inflammatory cytokines, such as IL-1β and IL-6, which are responsible for the sensitization of trigeminal afferent nerves. Studies have suggested that B-cell follicles can cause migraine attacks. However, they are more often detected in patients with progressive MS, which can explain the more frequent occurrence of migraine in these patients [36]. There is a correlation between MS, migraine, and inflammation. In MS patients with migraine, higher high-sensitivity C-reactive protein (hs-CRP) levels were detected compared to patients without migraine. Even though such a marker is not very specific, it illustrates the importance of inflammation in the coexistence of these two diseases. Higher levels of CRP were associated with the emergence and progression of periventricular and subcortical white matter hyperintensities on MRI scans [44]. These conditions may be correlated with the presence of small lesions in the cortical or midbrain [45]. Brain imaging studies show similar white matter abnormalities in patients with MS and migraine [46]. A disturbance in the antioxidant system of the body may be a contributing factor in the coexistence of MS and migraine [47]. The first-line defense mechanisms against free radicals are comprised of superoxide dismutase (SOD), catalase (CAT), and glutathione peroxidase (GSH-Px) [48]. Patients with RRMS who did not experience migraine exhibited lower levels of SOD, CAT, and GSH-Px compared to their healthy counterparts [47]. Moreover, the values of SOD, CAT, and GSH-Px in the patients with RRMS who experienced migraine were significantly lower than those in the patients without migraine [48]. The higher frequency of SOD and CAT polymorphisms in individuals suffering from migraine results in decreased antioxidant protection [49]. The RRMS patients without migraine had lower oxidant and antioxidant values than the healthy controls [47]. Furthermore, the antioxidant and oxidant values of the RRMS patients with migraine were significantly worse than those of the RRMS patients without migraine [47]. The formation and persistence of MS lesions are caused by reactive oxygen species and oxidative stress [50]. Many studies have demonstrated that migraine headache prevalence ranges from 8.9% to 41% of patients with MS (Table 1).

Migraine without aura is the most common form of migraine in adults and children, accounting for 70–80% of cases. The diagnosis is based on the ICHD-3 criteria and at least five attacks are required. Each attack lasts 4–72 h and is characterized by the following symptoms: unilateral location, pulsating quality, and moderate or severe pain intensity. Headache can be accompanied by nausea, photo- or phonophobia, and is exacerbated by physical activity [51].

Migraine with aura occurs in about 33% of patients with migraine. The diagnostic criteria include two attacks of fully reversible, gradually developing focal symptoms lasting from 5 to 60 min, followed by headache and other migraine symptoms. The aura includes visual, motor, sensory, brainstem, or speech symptoms [51].

The percentage of migraine with aura was higher only in one study in which two types of migraine (with and without aura) were considered. Rościszewska-Żukowska et al. showed that 41% of patients had migraine, of whom approximately 50% presented with migraine with aura and 30% of subjects were affected with migraine without aura. Probable migraine without aura was found in 23% of these patients [21]. Other studies indicated a higher percentage of patients with migraine without aura. 

Although migraine is linked to an increased risk of MS, the absolute risk difference between migraineurs and non-migraineurs is low. Migraine is identified as a potential prognostic factor for MS or as a risk factor contributing to the development of the disease. Some migraine patients may be misdiagnosed as having MS, although none of the theories has been conclusively confirmed [52].

### 6.2. Chronic Migraine

Chronic migraine is a type of headache occurring on 15 or more days a month for more than three months. In addition, the headache has the features of a migraine headache with or without aura on at least eight days per month [51].

Probable migraine is defined as the attacks missing one of the features that are required to fulfill all criteria for migraine headache and not fulfilling the criteria for another headache disorder [51]. Studies have demonstrated that the occurrence of probable migraine without aura ranges from 2.3% to 33% of MS patients (Table 1).

### 6.3. Tension-Type Headache

Tension-type headache (TTH) is the most frequent type of headache in the general population with an estimated global prevalence of 26% [37]. A typical attack lasts from 30 min to 7 days. It is of mild to moderate intensity, is located bilaterally, and has a tightening or pressing quality. As opposed to migraine, it is not exacerbated by physical activity, and it is not accompanied by nausea and vomiting, although photophobia or phonophobia may occur. According to ICHD-3, tension-type headache is divided into chronic and episodic types. Chronic TTH occurs 15 or more days a month for more than three months [51]. Many studies have confirmed that the occurrence of TTH ranges from 5% to 37.2% of MS patients (Table 1). Vacca et al. indicated that episodic TTH occurred in 6.3% of MS patients with headache and chronic TTH occurred in 0.8% of patients [16].

### 6.4. Cluster Headache

Cluster headache consists of severe attacks of unilateral pain in one or more regions (orbital, supraorbital, and temporal). It lasts for 15 to 180 min and occurs from once every other day to eight times a day and is accompanied by at least one of the following ipsilateral symptoms: conjunctival injection, lacrimation, eyelid edema, miosis, ptosis, facial sweating, and nasal congestion. Patients may also present with restlessness or agitation [51]. Cluster headache was diagnosed in one study with a low prevalence (0.8%). This type of headache is generally thought to have no significant correlation with MS [14,53]. 

### 6.5. Medication-Overuse Headache

Medication-overuse headache (MOH) occurs on 15 or more days a month due to the overuse of headache medication by patients with a primary headache disorder [51]. MOH affects 26% of MS patients [10].

## 7. Risk of Misdiagnosis

Headache is still considered a “red flag” in the diagnostic process, indicating potential alternative diagnoses to MS [54]. In their multicenter study, Solomon et al. assessed 110 patients misdiagnosed with MS and found that migraine alone or in combination with additional diagnoses was the most common alternative diagnosis (22% of cases). They argued that in those patients, migraine symptoms mistaken for demyelinating attacks were incorrectly used to satisfy dissemination in time criteria. Presumed migraine-associated white matter lesions were used to document dissemination in space criteria [55]. Another longitudinal study with 695 patients examined possible alternative diagnoses to MS. The subjects were included based on sensory, visual, brainstem, cerebellar, motor, cortical, or multifocal symptoms suggestive of MS [56]. They were given a full diagnostic workup and were followed up for three years. The MRI white matter changes were assessed for typical MS characteristics, i.e., oval morphology, diameter > 6 mm, and asymmetric situation in one of the following locations—perpendicular to the ventricles, periventricular, juxtacortical, infratentorial, or in the spinal cord. Should a given lesion not meet the above criteria, it was classified as atypical. By the end of the follow-up, migraine with atypical MRI lesions was the second most frequent alternative diagnosis. The study did not examine the effects of the use of DMTs on white matter changes [56].

The emerging data highlight the importance of a thorough differential diagnostic process, given the existing overlap in some of the symptoms and MRI findings in MS and migraine (particularly with aura), which may prove misleading to many clinicians and may often result in a misdiagnosis. As a result, misdiagnosed patients may be at risk of potential adverse drug reactions related to the unnecessary treatment of MS with DMTs. 

**Table 1 medicina-60-00572-t001:** The overview of studies investigating headache in multiple sclerosis.

Study	Study Group Characteristics	Prevalence of Headache in the MS Group	Headache Types in the MS Group
Vacca et al. Multiple sclerosis and headache co-morbidity. A case-control study [16]	N: 238 Age: 24–61 Both genders	Headache: 122 of 238 (51.26%) Headache in the prodromal phase: 85 of 122 (69.67%)	MO: 31.9% MA: 3.8% ETTH: 6.3% CTTH: 0.8% PMA: 6.3%
Rolak et al. Headaches and multiple sclerosis: a clinical study and review of the literature [18]	N: 104 Age: 22–62 Both genders	Headache: 54 of 104 (52%) Headache in the prodromal phase: 7 of 54 (12.96%)	MH: 21% TTH: 31%
Beckmann and Türe Headache characteristics in multiple sclerosis [10]	N: 754 Age: 15–61 Both genders	Headache: 515 of 754 (68.3%) Headache in the prodromal phase: 101 of 515 (20%)	MO: 17% MA: 6.9% PMA: 2.3% CM: 0.7% TTH: 20% MOH: 38% UH: 2%
Putzki et al. Prevalence of migraine, tension-type headache and trigeminal neuralgia in multiple sclerosis [57]	N: 491 Mean age: 45.3 Both genders	Headache: 295 of 491 (62.5%) Headache in the prodromal phase: NI	TTH 37.2% MH: 13.5% PMA: 5.1% MH and TTH: 8.1%
D’Amico et al. Prevalence of primary headaches in people with multiple sclerosis [53]	N: 116 Mean age: 41.3 Both genders	Headache: 67 of 116 (57.7%) Headache in the prodromal phase: NI	MH: 25% TTH: 31.9% CH: 0.8%
Demetgul et al. Investigation of the Association between Headache Type, Frequency, and Clinical and Radiological Findings in Patients with Multiple Sclerosis [34]	N: 320 Age: 17–68 Both genders	Headache: 174 of 320 (54.4%) Headache in the prodromal phase: NI	MH: 30.06% MO: 16.38% TTH: 23.80%
Rościszewska-Żukowska et al. Clinical Characteristics of Headache in Multiple Sclerosis Patients: A Cross-Sectional Study [21]	N: 419 Age: 18–71 Both genders	Headache: 236 of 419 (56%) Headache in the prodromal phase: 186 of 236 (78.8%)	MH: 41% MA: 45% MO: 30% PMA: 23% TTH: 14%
Terlizzi et al. Headache in multiple sclerosis: prevalence and clinical features in a case control-study [35]	N: 150 Mean age: 40 Both genders	Headache: 80 of 150 (53.3%) Headache in the prodromal phase: NI	MH: 31.33% TTH: 14%
Möhrke et al. Headaches in multiple sclerosis patients might imply an inflammatorial process [58]	N: 180 Mean age: 43.9 Both genders	Headache: 98 of 180 (55.4%) Headache in the prodromal phase: NI	MH: 8.9% MA: 1.1% MO: 7.8% TTH: 12.8% UH: 32.78%
Nicoletti et al. Headache and Multiple Sclerosis: A Population-Based Case-Control Study in Catania, Sicily [59]	N: 101 Mean age: 43.6 Both genders	Headache: 58 of 101 (57.4%) Headache in the prodromal phase: NI	MH: 19.8% TTH: 23.8% UH: 5.9%
Gebhardt et al. Headache at the Time of First Symptom Manifestation of Multiple Sclerosis: A Prospective, Longitudinal Study [14]	N: 50 Mean age: 30 Both genders	Headache: 78% initially, 61% after 6 months Headache in the prodromal phase: NI	MH: 14% PMA: 33% TTH: 14% UH: 8%
Özer et al. Headache in multiple sclerosis from a different perspective [60]	N: 100 Mean age: 33.9 Both genders	Headache: 68 of 100 (68%) Headache in the prodromal phase: NI	MH: 14% TTH: 16% TTH+MH: 8% PSH: 11% PSH+MH: 6% MOH: 6% UH: 1%
Gustavsen et al. Migraine and frequent tension-type headache are not associated with multiple sclerosis in a Norwegian case-control study [61]	N: 510 Mean age: 50.7 Both genders	Headache: 52.4% Headache in the prodromal phase: NI	MH: 18.2% TTH: 12.7% Control: *N* = 914 Age = 43.9 Headache: 54.4% MH: 16.3% TTH:14.9%
Sahai-Srivastava et al. Headaches in multiple sclerosis: Cross-sectional study of a multiethnic population [62]	N: 233 Mean age: 44.4 Both genders	Headache: 115 of 233 (49.4%) Headache in the prodromal phase: NI	MH: 36% TTH: 5%
Kister et al. Migraine is comorbid with multiple sclerosis and associated with a more symptomatic MS course [63]	N: 204 Mean age: 45 Both genders	Headache: 131 of 204 (64.2%) Headache in the prodromal phase: NI	MH: 46.1% TTH: 18.1%
Doi et al. Frequency of chronic headaches in Japanese patients with multiple sclerosis: with special reference to opticospinal and common forms of multiple sclerosis [17]	N: 127 Mean age: 45.1 Both genders	Headache: 64 of 127 (50.4%) Headache in the prodromal phase: 28 of 127 (22%)	MA: 3.9% MO: 16.5% ETTH: 27.6% CTTH: 2.4% MOH: 0.8% UH: 1.6%
Villani et al. Primary headache and multiple sclerosis: preliminary results of a prospective study [64]	N: 102 Age: 15–68 Both genders	Headache: 61.8% Headache in the prodromal phase: On average, headache preceded MS onset by 8 years.	MH: 45.1% TTH: 8.8%
Gee et al. The association of brainstem lesions with migraine-like headache: an imaging study of multiple sclerosis [39]	N: 277 Age: NI Both genders	Headache: 154 of 277 (55.6%) Headache in the prodromal phase: NI	MH: 34.3% TTH: 14.1% MH+TTH: 7.2%

MH—migraine headache; MO—migraine without aura; MA—migraine with aura; NI—no information; CM—chronic migraine; TTH—tension-type headache; ETTH—episodic tension-type headache; CTTH—chronic tension-type headache; PMA—probable migraine without aura; CH—cluster headache; MOH—medication-overuse headache; PSH—primary stabbing headache; UH—unclassified headache.

## 8. Role of Disease-Modifying Therapies

Disease-modifying therapies (DMTs) for MS can potentially exacerbate or induce headache. In MS patients, headache consultations are prevalent with 89% of patients seeking medical advice. The analysis of the data indicated a higher rate of physician consultation in patients experiencing MOH (78%) compared to other patient groups with migraine (52%) or TTH (46%) [10].

Following the diagnosis of MS, headache may occur as a side effect of DMTs. Beckmann and Türe found that interferon-beta (IFN-β) was associated with a higher incidence of headache, particularly when initiated early and when treatment was shorter [10]. Treatment with IFN-β can cause headache accompanied by flu-like symptoms with an incidence of 67% in subjects compared to the placebo group (57%) [28,65,66]. New-onset headache occurred in 70% of patients treated with IFN-β, while 50% reported the exacerbation of headache [67]. In the CAMMS223 trial comparing alemtuzumab with IFN-β-1a, 55% of alemtuzumab-treated patients and 66% of interferon-treated patients reported headache mainly due to infusion reactions [68]. Doi et al. found that MS patients treated with IFN-β experienced headache more frequently than those who were not on the therapy (42.4% vs. 23.4%) [17]. 

Medication-overuse headache was more common in patients treated with teriflunomide [10]. Infusions with ocrelizumab and daclizumab were not related to the occurrence of headache in the OPERA I and II or the DECIDE clinical trials [69,70]. Natalizumab demonstrated no statistically significant association with any type of headache. In the female group, there was no correlation with treatment type [14]. In female patients, migraine is more commonly reported during interferon beta treatment. Patients with migraine also have a longer history of MS medication use prior to headache onset compared with various headache types [14]. TTH is more prevalent than other headache types after fingolimod treatment [14]. In a 2022 meta-analysis, the relative risk of headaches during B-cell targeted therapy was 12% higher, but it was not statistically significant. This study included cladribine, ocrelizumab, ofatumumab, and rituximab. Only the effect of cladribine on increasing the frequency of headaches was statistically significant [71]. 

According to a network meta-analysis, patients with relapsing–remitting MS (RRMS) receiving alemtuzumab reported headache more frequently than those on placebo or other highly effective DMTs. Additionally, patients on cladribine reported headache more frequently than those on natalizumab. Headache was also more frequent in those on fingolimod versus natalizumab [12]. In turn, Togha et al. found that individuals treated with INF-1a and INF-1b reported more headaches as opposed to patients on fingolimod or glatiramer acetate [27].

Headache was one of the most common adverse effects of therapy in 24% of patients [18]. The risk of headache was significantly increased with laquinimod therapy [11].

None of the studies reported any improvement in headache episodes following the use of DMTs [10,11,12,17,27,28,65,66,67,68,69,70,71,72].

## 9. Therapy of Headache in MS

The management of headache in MS follows the same approaches as in the general population, including acute symptom control and daily prophylactic medications to decrease headache frequency [28]. The guidelines for preventive medication use were outlined by the American Migraine Prevalence and Prevention study. They recommend daily preventive treatment if there are four or more headache days with at least some impairment, three or more headache days with severe impairment, or requiring bed rest with six or more headache days per month [73]. Commonly prescribed drugs for migraine prevention include angiotensin-converting enzyme inhibitors, beta-adrenergic blockers, calcium channel antagonists, antiepileptics, and antidepressants [74]. Treatment principles for acute migraine attacks involve early intervention and the use of non-specific or specific agents. Non-specific treatment includes antiemetics, neuroleptics, and nonsteroidal anti-inflammatory drugs, while specific agents, such as dihydroergotamine and triptans selectively activating 5-HT receptors, are used in acute migraine attacks [75,76]. Caution is warranted in managing migraine headache in MS patients due to an increased risk of adverse reactions related to potential drug interactions [14].

In their prospective longitudinal study, Gebhardt et al. obtained satisfactory results by treating 39 patients with high-dose intravenous methylprednisolone for five days, with 74.4% of subjects reporting partial improvement or complete headache remission [14].

Management of headache in MS follows the same approaches as in the general population with attention paid to acute symptom control and prophylactic medication use based on the guidelines. Caution is advised due to potential adverse reactions.

Preventive therapies should be approached with caution due to the increased risk of adverse events associated with multiple drug interactions [77]. Although tricyclic antidepressants are one of the most prescribed treatments for migraine in adults, they should be used with caution in some patients with MS. Tricyclics can aggravate urinary dysfunction (episodes of urinary retention), constipation, and MS-related cognitive impairment. However, a low dose of tricyclics is advised to ensure efficacy with minimal or no side effects [78]. Anti-CGRP medications are both safe and effective for migraine treatment. Nonetheless, it is unclear if its combination with DMTs in MS patients results in increased adverse events or disease activity. Many CNS cells, including astrocytes, oligodendrocytes, and microglia, have CGRP receptors and are partially affected by CGRP, resulting in reduced neuroinflammation in experimental and animal models of MS. However, when nociceptive signaling activates microglia, it can cause inflammation in pain pathways [79]. The use of sodium valproate or valproic acid as a migraine prophylaxis in young women of childbearing age with MS is limited due to associated side effects such as polycystic ovaries and the well-known teratogenic potential. Topiramate has a similar concern; there is a higher risk of cleft lip and/or cleft palate, which is a contraindication for pregnant women to use this medication [43].

## 10. New Perspectives in the Treatment of Headaches in Patients with MS

Anti-CGRP monoclonal antibodies (erenumab, eptinezumab, galcanezumab, fremanezumab) seem to be a new, promising option for patients with migraine. There are limited data about their use in individuals with MS [36]. Gonzalez-Martinez et al. conducted a study on nine MS patients with migraine receiving both DMTs and anti-CGRP monoclonal antibodies. They have not noted any drug interactions or MS progression during the observation period. Both therapies were well tolerated [80]. Despite the promising outcomes, the study group was too small. We need more studies to assess the safety and efficacy of such combination therapy. 

Another option for patients with chronic migraine is treatment with botulinum toxin. In MS patients botulinum toxin is also used to reduce the spasticity symptoms. In this case, we need to bear in mind the possibility of neutralizing antibody formation. Patients should be administered adjusted doses at appropriate intervals and in different locations. However, due to its high safety profile and high effectiveness, botulinum toxin is the recommended treatment for patients with chronic migraine and MS [36].

## 11. Conclusions

In conclusion, the complex interplay between headache and MS underscores the need for a nuanced approach to diagnosis and management. As a potential prodromal symptom, headache is significant not only in the early phases of MS but also during the disease course, thus impacting the quality of life in MS patients. Headaches are a major issue for multiple sclerosis patients. They are related to a decline in quality of life compared to people with MS without headaches, as well as more frequent tiredness or depression in patients with migraines and MS. The most prevalent type of headache seen in MS patients is migraine without aura. Headaches are roughly 2–3 times more prevalent in women than in males, and those without headaches have a lower average age than those suffering from them. The prevalence of headache in MS patients, particularly migraine and tension-type headaches, should be taken into consideration in differential diagnosis to avoid potential misdiagnosis and the associated risks. DMTs for MS may further influence headache patterns, emphasizing the importance of personalized treatment plans and close monitoring. As the understanding of the intricate relationship between headache and MS evolves, healthcare practitioners should remain vigilant, adopting a comprehensive and patient-specific approach to address this multifaceted aspect of MS care.

## Figures and Tables

**Figure 1 medicina-60-00572-f001:**
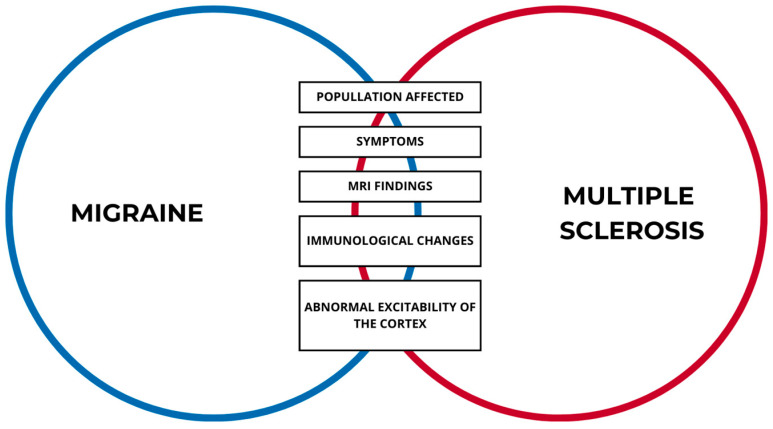
The similarities between MS and migraine headache pathogeneses.

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
