# Peer review of "Headache in Multiple Sclerosis: A Narrative Review"

_medicina, 2024, doi:10.3390/medicina60040572_

Round 1

Reviewer 1 Report (Previous Reviewer 2)

Comments and Suggestions for Authors

Adamczyk et al. narratively reviewed the literature regarding headaches and multiple sclerosis. I appreciate the corrections from the last review, but I would only like to highlight some other recommendations.

1.     Please provide a chapter about new perspectives

2.     Remove the citations in the conclusion.

3.     Consider reviewing the reference list. There are 88 articles cited, and the reference list currently has 69 articles.

Author Response

Dear Reviewer 1,

Thank you very much for your second revision. We really appreciate your effort to improve our manuscript. We have corrected the paper according to your suggestions. The new changes are highlighted in purple color.

1. We have added a chapter about a new perspectives.

9. New perspectives in the treatment of headaches in patients with MS

Anti-CGRP monoclonal antibodies (erenumab, eptinezumab, galcanezumab, fremanezumab) seem to be a new, promising option for patients with migraine. There is limited data about their use in MS individuals (41). Gonzalez-Martinez et al. conducted a study on 9 MS patients with migraine receiving both DMTs and anti-CGRP monoclonal antibodies. They have not noted any drugs interactions or MS progression during the observation period. Both therapies were well tollerated (89). Despite the promising outcomes, the study group was too small. We need more studies to assess safety and efficacy of such combination therapy.

Another option for patients with chronic migraine is treatment with botulinum toxin. In MS patients botulinum toxin is also used to reduce the spasticity symptoms. In this case we need to bear in mind the possibility of neutralizing antibodies formation. Patients should be administered adjusted doses at appropriate intervals and in different locations. However, due to its high safety profile and high effectiveness, botulinum toxin is the recommended treatment for patients with chronic migraine and MS (41).

2. Thank you for this remark, we removed the citations in the conclusion.

3. We are sorry for this mistake. We have reviewed the reference list and updated them.

We are grateful for your valuable remarks on our paper. We truly hope that that all changes were made in accordance to your suggestions.

Sincerely,

Bożena Adamczyk

Reviewer 2 Report (Previous Reviewer 3)

Comments and Suggestions for Authors

The authors made the corrections from previous review, therefore I recommend the manuscript for publication.

Author Response

Dear Reviewer 2,

Thank you very much for your second revision. We are grateful for your valuable remarks. We really appreciate your effort to improve our manuscript. 

Sincerely,

Bożena Adamczyk

This manuscript is a resubmission of an earlier submission. The following is a list of the peer review reports and author responses from that submission.

Round 1

Reviewer 1 Report

Comments and Suggestions for Authors

The article is well-structured, providing detailed information on the various assessments conducted. Implementing these suggestions can enhance the clarity, conciseness, and overall accessibility of the article, ensuring that readers can quickly grasp the key insights.

1.       Label each section in the abstract with informative headings to enhance the organisation and readability of the content.

2.       B-Cell Targeted Therapy- If available, provide additional information or context regarding the observed trend related to B-cell targeted therapy.

3.       Consider breaking down the introduction into shorter paragraphs for better flow.

4.       Emphasise specific challenges or considerations in managing headaches in MS patients, mainly when using preventive medications.

5.       Clearly distinguish between various headache types induced by different disease-modifying therapies to aid comprehension.

6.       The section on the role of disease-modifying therapies is detailed but could benefit from a concise summarisation of critical findings in the introduction.

7.       Expand on the caution about drug interactions, specifying potential risks and highlighting specific drugs or classes that may pose challenges.

8.       Consider briefly addressing the findings' global relevance to widen the article's scope.

9.       Clarify the significance of the prodromal phase and its relevance to the overall understanding of MS.

10.   Consider mentioning if there are ongoing efforts to establish formal criteria for diagnosing prodromal MS.

11.   Elaborate on how headaches, especially migraines, affect the quality of life for MS patients, possibly with some specific examples or anecdotes.

12.   Offer a concise summary of the observed gender and age differences in the occurrence and impact of headaches in MS patients.

13.   Consider providing concise definitions or characteristics of each type (e.g., migraine, tension-type headache) to assist readers in understanding the distinctions.

14.   The introduction provides a comprehensive overview of MS, its prevalence in Poland, and the difficulty in predicting disease progression. Including specific numbers, such as the estimated affected individuals and prevalence, adds quantitative context.

15.   The introduction is well-supported with citations, reinforcing the credibility of the information presented. However, it might be helpful to include the publication year in the references for clarity.

16.   Ensure consistency in the use of terminology. For example, in the introduction, both "moderate symptoms" and "severe changes" are mentioned without clear definitions. Providing clarity on these terms would improve understanding.

17.   Ensure that references are consistently formatted and up-to-date, providing a credible foundation for the information presented.

18.   Provide a brief explanation or definition of Radiologically Isolated Syndrome (RIS) for readers who may not be familiar with the term.

19.   Briefly mention the methodologies employed in the studies cited, offering readers insight into the strength of the evidence presented.

Comments on the Quality of English Language

The language is generally clear, but in some sections, especially in the introduction, there is room for simplification without sacrificing the depth of information.

Reviewer 2 Report

Comments and Suggestions for Authors

1. The type of literature review should be described in the title.

2. L40-43, provide a citation. The information is specific.

3. If no formal criteria were made for the prodromal phase of MS, please provide one description used in one of the original article studies for this manifestation. Or a table comparing the different definitions of prodrome.

4. Provide the methodological design of the present study. A small chapter can be provided showing databases and search terms.

5. Regarding “Chapter 7,” did any study show improvement in headache episodes with DMTs? Or was it only observed worsening? How could the authors explain these findings?

6. How were confounding variables controlled throughout the studies in the literature assessing headaches in MS patients?

7. How did the studies assess white matter changes associated with headaches? How were these findings evaluated in MS individuals? Did the use of DMTs improve the white matter changes?

8. Could the authors provide a figure or schematic diagram explaining the relationship between these two disorders? This would significantly impact the quality of the manuscript.

9. The reference list should be reviewed. Some references need to be authors added, and others have different styles.

Reviewer 3 Report

Comments and Suggestions for Authors

The manuscript entitled Headache in multiple sclerosis, brings a new inside in the diagnosis and potential treatments for multiple sclerosis symptoms.

Observation

Introduction

- line 30-33 - please indicate all MS clinical forma.

- line 40 - indicate the references here.

- at the end of introduction please write more details about your study, what is useful for, and what new perspectives in bringing regarding the diagnosis and treatments of MS.

Chapter 3 

A table with your findings would be more suggestive for pointing the headache as the first symptom of MS.

Same for Chapter 4.

- line 164 - please indicate the references here.

The assumption the migraine and MS has a relationship must be documented!!

- line 168 - please offer more details about pathological cascade in this paragraph.

- line 173 - explain cortical MS.

Please offer more details about the pathogenetic processes in migraine, that can be similar in MS.

The table 1 is to long, please split it and organize it it for each chapter above. 

Conclusion are to general, you you should refer more specific about your study.